# Perspectives and experiences of researchers regarding feedback of incidental genomic research findings: A qualitative study

Joseph Ochieng[1]*, Betty Kwagala[2], John Barugahare[3], Erisa Mwaka[1], Deborah Ekusai-Sebatta[1], Joseph Ali[4,5], Nelson K. Sewankambo[6]

**1** Makerere University School of Biomedical Sciences, Kampala, Uganda, **2** Makerere University School of Business and Management Studies, Kampala, Uganda, **3** Makerere University School of Liberal and Performing Arts, Kampala, Uganda, **4** Johns Hopkins Bloomberg School of Public Health, Baltimore, MD, United States of America, **5** Johns Hopkins Berman Institute of Bioethics, Baltimore, MD, United States of America, **6** Makerere University School of Medicine, Kampala, Uganda

* ochienghoe@yahoo.com

## Abstract

### Background

There is a plethora of unanswered ethical questions about sharing incidental findings in genetics and genomics research. Yet understanding and addressing such issues is necessary for communicating incidental findings with participants. We explored researchers' perspectives and experiences regarding feedback of incidental genomics findings to participants.

### Methods

This was a qualitative study using semi-structured interview schedules for In-depth interviews. Thirty respondents were purposively selected based on role as genetics and genomics researchers in Uganda. Data were analysed through content analysis to identify emerging themes using a comprehensive thematic matrix. QSR International NVivo software was used to support data analysis.

### Results

a). On perceptions, sharing of incidental findings was acceptable and four themes emerged including role of professional judgement; role of ethics committees and ethical guidelines; optimal disclosure practices; limits to professional duty and uncertainty and; b). on practices, sharing had been carried out by some researchers and a theme on experience and practices emerged.

### Conclusion

Feedback of incidental genomics research findings to participants is generally acceptable to researchers. Some researchers. Challenges include lack of ethical guidelines and uncertainty about the findings.

**Data Availability Statement:** All relevant data are within the manuscript and its attached Supporting Information files.

**Funding:** Research reported in this publication was supported by the National Human Genome Research Institute of the National Institutes of Health under Award Number U01HG009822. The content is solely the responsibility of the authors and does not necessarily represent the official views of the National Institutes of Health. The funders had no role in study design, data collection and analysis, decision to publish, or preparation of the manuscript.

**Competing interests:** The authors have declared that no competing interests exist.

**Abbreviations:** GGR, Genetics and Genomics Research; H3Africa, The Human Heredity and Health in Africa (**H3Africa**) Initiative; IF, Incidental Findings; REC, Research Ethics Committee; UNCST, Uganda National Council for Science and Technology.

## Introduction

Although genetics and genomics research (GGR) is expected to be a public good that would contribute to improved health care and disease outcomes, a number of ethical, social and regulatory issues associated with this research remain unresolved. Aspects such as return of genetic information derived during research continue to be an issue globally and a topic of intense debate [1–3]. Although human GGR can yield information that may be of clinical relevance to the individuals who participate in research, there is not yet a consensus on the responsibilities of researchers to disclose individual research results to research participants [4, 5]. In many settings, there is no legal obligations for researchers to return genetic results to participants despite patient willingness to receive the results [6, 7]. Additionally, GGR testing has a potential for identifying incidental or secondary findings unrelated to the objective of the testing but of potential medical value for the individual being tested. Incidental findings (IFs) have been used in a variety of clinical and research contexts to indicate unexpected positive findings [8]. Incidental findings have further been defined as findings having potential health or reproductive importance for an individual, discovered during genetics and genomics testing but beyond the aims of the testing [7, 9]. Such incidental findings have been reported to occur at a rate of 0.9% to 8.8% in some populations [10].

While the global debate on IFs is taking shape, literature on the African perspective, Uganda inclusive, is still limited [1, 11]. Yet, African genomes have been characterized as having more variants than in any other population in the world. Thus, the need to address issues such as whether to, how and which IFs to share following testing in African populations [12].

The debate about the circumstances under which researchers are obliged to feedback IFs is ongoing with suggestions that such determination be made at an institutional level, on a case-by-case basis often by Research Ethics Committees (RECs). However, in many countries, nationally accepted or contextualized guidelines to direct researchers' or REC members' decisions about disclosure of IFs discovered during GGR and testing do not exist [13, 14].

In Uganda, the available national guidelines for conduct of research involving humans are silent about GGR including the sharing of IFs [15]. International guidelines such as those drafted by the H3Africa consortium stress clinical validity and relevance for the target population as important considerations [16]. However, determination of clinical validity and relevancy is often left to the discretion of researchers or attending doctors, leaving room for a significant degree of idiosyncrasy.

In the absence of specific guidelines, many researchers argue that only IFs with confirmed clinical utility; with a possibility of treatment or prevention should be disclosed [10, 14]. However, for many countries uncertainty remains regarding whether, when, how and to whom to offer IFs [6]. Other concerns include the relevance of IFs to the participants' or patients' family members and if researchers and clinicians should regard genetic information as confidential to individuals or to families [17]. This is because it's still unresolved in many settings on whether clinicians/researchers have an obligation to warn family members of potential risk associate with a genetic disorder identified in a family member when such an individual is not willing to disclose their genetic information.

Since literature and debate on GGR and associated incidental findings has been evolving over the last decade, an updated discussion on this topical issue is necessary to provide updated information on the subject matter. This study explored researchers' perspectives and experiences regarding feedback of incidental genetics and genomics research findings to participants.

## Methods

This was a cross-sectional study that employed a qualitative exploratory approach. The study was conducted by a team of academics comprising of social scientists, bioethicists and medical scientists including both male and females with experience in qualitative research. Data was collected between 2019 and 2020 among researchers who were actively involved in GGR in Uganda. Respondents were recruited from institutions in the Central, Western, Eastern and Northern parts of the country. Data were collected using in-depth interviews that were guided by a semi-structured interview schedule adapted from a checklist for genetic and genomic testing of the Federal Code of Regulations which provides detailed information on the requirements for research participants to be respected throughout the research process [18]. Thirty researchers were identified with help of one of the leading GGR researcher in the country and purposively selected based on their being investigators on GGR projects.

Data collection entailed face to face in-depth interviews conducted in the English language privately conducted at the offices of the respondent and lasted 40–70 minutes. Responses were audio recorded which was complemented by notes taken by a research assistant. Recorded information was transcribed verbatim and checked for accuracy before analysis. Data were analysed through content analysis along the main themes of the study. Content analysis was conducted using a comprehensive thematic matrix that included codes, categories and themes to identify common patterns arising from the narratives. The coding was done both deductively and inductively. Transcripts were further reviewed for emerging themes which were integrated into the thematic matrix. Multiple people JO, BK, DES were involved in applying and confirming application of codes across all transcripts and disagreements were resolved by cross checking with the recorded data. NVivo software (QSR international 2020) was used to support data analysis and illustrative quotes were extracted.

### Ethical considerations

Ethical review and approval was obtained from the Makerere University School of Biomedical Sciences Higher Degrees and Research Ethics Committee ref. SBS 628, followed by clearance by the Uganda National Council for Science and Technology (UNCST) ref. SS268ES.

Researchers who had provided written informed consent participated in the study. No participant identifying information was recorded.

## Results

Of the 30 study respondents, 23 were male. The mean age was 41 years (range, 29–65years). All were residents of Uganda and affiliated to one of five research institutions in the Central, Western, Eastern and Northern parts of the country. The participants' main job roles included academic and researcher roles with most 24/30 (80%) having at least five years of GGR experience (see Table 1).

Fields of specialization included molecular biology, immunology, microbiology, biochemistry, pharmacology, internal medicine, transfusion medicine, surgery, obstetrics, and gynaecology. Types of studies conducted by the respondents included molecular diagnostics, pharmaco-genetics, pharmaco-genomics, molecular genotyping, microbio-genotyping and haematological genomics. Content analysis identified five themes; a). On perspectives, sharing of incidental findings was acceptable and four themes emerged including; 1). role of professional judgement; 2). role of ethics committees and ethical guidelines; 3). optimal disclosure practices; 4). limits to professional duty and uncertainty and; b). on experiences, sharing had been carried out by some researchers and a theme on 1). experience emerged.

 

**Table 1. Demographic characteristics of respondents.**

| Attribute | No of participants N = 30 | Male | Female |
|---|---|---|---|
| **Age range** | | | |
| Missing age | 1 | | 1 |
| 20–29 | 1 | 1 | 0 |
| 30–39 | 10 | 8 | 2 |
| 40–49 | 10 | 8 | 2 |
| 50–59 | 6 | 6 | 0 |
| 60–69 | 2 | 0 | 2 |
| **Education level** | | | |
| Masters | 9 | 8 | 1 |
| PhD | 21 | 15 | 6 |
| **Employment/ Position** | | | |
| Researcher | 6 | 5 | 1 |
| Lab associate | 3 | 3 | 0 |
| Dean | 2 | 1 | 1 |
| Lecturer | 11 | 10 | 1 |
| Professor | 4 | 2 | 2 |
| Senior scientist | 1 | 0 | 1 |
| Director | 2 | 1 | 1 |
| Graduate fellow | 1 | 1 | 0 |
| **Duration/experience in genomics research** | | | |
| 1–4 years | 6 | 5 | 1 |
| 5–10 years | 15 | 13 | 2 |
| 11–15 years | 5 | 3 | 2 |
| >15 years | 4 | 2 | 2 |

Note: Some of the respondents held multiple roles and positions

## 1. Role of professional judgment

Many respondents had a positive attitude towards sharing incidental GGR findings, particularly findings judged to have potential clinical significance. It was noted that incidental findings usually come up during screening and exploration but despite the lack of consent researchers should use their judgement and should have the responsibility to share the information with the participant because they are dealing with human life and human beings. Respondents stressed the fact that researchers as professionals are obliged to communicate incidental findings to participants. Some of the respondents suggested that researchers should exercise their discretion about whether to share certain incidental findings: As observed by the following quotes

> "I think these are findings that you didn't expect but then turn out as relevant. But if the findings have impact on the treatment of this participant and clinically relevant, then I think I would first approach the attending doctor because we enrolled theses participants from the Hospital, so I would first discuss with the attending doctor and discuss the results and then from there we can take it on." R029

Respondents felt that dealing with incidental findings was a difficult process since the results are not anticipated. Some respondents noted that sharing of incidental findings

requires not only caution but also adherence to institutional procedures for reporting such findings. Others observed that in the absence of contextualized ethical guidelines, regulatory and donor opinion should be sought.

*"Like I said, you just have to go through the procedure of getting back, because some of incidental findings could mean a life and death situation. . . you have to go through the procedures of the hospital where you obtained the samples and inform them. . . .."* R017

Furthermore, some respondents noted that it is important to only consider results that are beneficial or actionable and that incidental findings should be returned to participants.

*"If a result is actionable it should be communicated because knowledge is power, you give this person the knowledge and then they have the ability to make a decision on when to act."* R015

Although many researchers considered it good practice to share incidental findings, on the contrary, some felt that some findings should not be shared if they are accidental and unexpected, or not actionable, Some of the respondents believe that the decision to share findings should be at the discretion of the researcher and the participants' expressed desires, especially in cases where the participant expressly indicates that they do not wish to know such results or where the results could cause more harm than good to associated persons. Hence, the need to consider the context or situation.

*"At the end of the day, I think the decision to return results is very contextual. If a participant just says I don't want to know' for example paternity identifying information."* R010

*"If it is not actionable then you leave it you forget about it."* R015

## 2. Role of ethics committees and ethical guidelines

Many respondents argued that sharing of incidental findings with the research participants is an ethical obligation. However, such communication should be guided by a research ethics committee either prior to recruitment of participants or if sharing of incidental findings was not part of the approved consent form, then the researcher has to seek clearance from the ethics committee before sharing such findings with the participants. It was noted that guidance from regulators is important since the current ethical guidelines lack clarity on this issue yet regulators might be in a better position to anticipate potential outcomes of sharing such findings.

*"I think it would be worthwhile to go back and discuss with the ethics bodies how such results would be disseminated because, this is not something you set out to do . . . these things (results) you found them incidentally. It would be better to go back to the ethics bodies and discuss how this information could be communicated because it could be beneficial to the individual and to the community."* R006

*"So incidental findings I think once you come across such information its important but since ethical research is determined and regulated by some policies, if it is not clear within the framework currently, I think it is best to may be discuss with the research regulators."* R002

Respondents believed that in order to share incidental GGR findings ethically, informed consent should be sought from the research participants prior to sharing of the findings Availability of ethical guidelines would help in addressing aspects such as participants' choice

regarding receipt of genetics and genomics results. Others felt that if sharing of incidental findings was completely outside what you set out to do, then there is need to consult both the ethics committee and the participant because it may involve review of the consent form. So, either way, participants participation either emotionally, socially and other wise is the first thing that you must consider.

> *"Participants can be informed that the technique to be used has the capability of getting more information than what is being tested for . . . therefore in that case if there is something useful, we will let you know. I think this can be included in the consent."* R009

> *"If it is already pre-introduced in the consent form and the Research Ethics Committee has already approved the study, then absolutely. Then you would be able to actually go back to the community and share."* R006

## 3. Optimal disclosure practices

Respondents suggested a number of best practices that should accompany feedback of incidental findings. While sharing incidental GGR findings was considered important, adequate preparation is essential given the potential sensitivity of such findings. Sharing of genetics and genomics results has been observed to cause anxiety and unnecessary suffering to the individuals and their families. It should be noted that genome-wide association studies have identified various strong associations between genetic polymorphisms and susceptibility to common infectious disease phenotypes, such as HIV-1, hepatitis B and C viruses, dengue, malaria, tuberculosis, leprosy, meningococcal disease and prion disease.

> *"When there is a possibility of detecting for certain infections or conditions that are of public concern, you might need to prepare these participants. Participant should be aware of such possibilities and this should be included in the consent form."* R013

Respondents stressed that incidental GGR findings should always be shared in the presence of a qualified genetic counsellor who could provide the appropriate psycho-social support to the participant and guard against any potential misconceptions about the implications of the results.

> "Counselling is *very very important, because of the possible outcomes. We should have genetic counsellors. Whenever you talk about genetics, there are a lot of misunderstandings. So presence of genetic counsellors would help sort out those possible misunderstandings."* R005

> *"I again being humane, I think I would report such a finding but the way to handle such also matters. I think you need a counsellor to explain if you are not counsellor yourself. Maybe you need a genetic counsellor or a counsellor who can break it down to the patient."* R029

Respondents observed that despite the necessity to communicate incidental findings, consideration of the implications of such findings to the individual participants, the family and the community needed to be evaluated. Respondents placed emphasis on the need to consider the nature of incidental findings such as the gravity of the finding (whether it is potentially fatal); and if the finding has public health implications. Possible risks and benefits associated with sharing the findings have to be evaluated. Otherwise it would not be good to share findings that would cause harm to research participants and their communities.

> *"Risks! It is important to establish the risks involved. The sharing of results has to be evaluated in light of their impact, prior to their communication.. . ."* R005

*"If the conditions allow I think I would share findings, because if their (participants') life will depend on that, or they are likely to improve their quality of life. I think its humane to communicate and if they can get treatment." R029*

Communicating incidental genomics findings was generally acceptable if the results are clinically valid and relevant to the target individual or population.

*"It still comes down to what is the benefit of giving these results to this person, some results are incidental but are so important that you need to share them with the person for them to be able to take action." R003*

*"Incidental findings are also very difficult at some point because you didn't go for them but you have found them. But as I said, it still comes down to what is the benefit of giving these results to this person, some results are incidental but are really important." R001*

Respondents noted that in case of public health concerns, the consent of participants should be waived; authorities have to be informed.

*"If a person has not consented but in conducting a whole genome sequence you find that this person actually has this condition that is a public health concern, you must inform not only the patient but also the authorities; in fact you must follow up this patient." R010*

Other respondents felt that incidental genetics and genomics research findings are like any other findings generated during the conduct of research. Sharing of results with research participants is encouraged as part of the feedback process. As such results that have meaning can be shared with the participants for any necessary action.

*"Incidental findings again to me I would treat them like any other findings, do they have meaning? Is that meaning beneficial?" R009*

## 4. Limits of professional duty and uncertainty

Respondents would prefer communicating incidental findings when they are certain of the results and after such tests have been verified by other researchers. Hence the need for caution, unless it is a life-threatening condition where one would risk communicating uncertain information to provide early warning to the affected individual. Thus the need for a peer review process so as to be sure of the findings before disclosure.

*"Yes. . .you will have to categorize the incidental finding, in terms of is it a life and death, fatal in that if this person doesn't know, they are at risk? When you get such findings, you also weigh the risks, and be sure that the findings are correct. . . .You need to know beyond reasonable doubt that if I don't tell this individual, this will happen." R017*

*"But, still my suggestion would be that before this kind of information is taken back to the communities, a very clear assessment of the impact should be established because genetics do not always lead to Phenotypic presentation. . . I think it's better to establish the actual impact". R006*

Respondents observed that for any results generated, there was need to further investigate so that one is sure of what they are sharing with the participants. Researchers should avoid rushing to share information which may turn out to be different in the future following further research.

*"Its good to investigate further to really get to the bottom before sharing. This is because you can learn of information and before you investigate it it's not good to run down with it, it's good to consult more and then together you might find the best way to handle it." R002*

## 5. Experiences

Although incidental findings have not been shared by most of the research studies, some studies have shared their findings with research participants. The procedures and practices for sharing the findings varied among the researchers: some described it as giving just information to participants, others provided the information to the attending clinicians, while others employed a comprehensive community engagement process. Similarly, researchers noted several challenges in the process including the lack of guidance due to inadequacy of existing ethical guidelines as observed below.

*"This is not a rare situation . . .. I have had to do this on a case-by-case basis for instance when we first discovered a child with sickle cell, a condition called sickle cell variant, O, it was so rare . . . When we first got that it was so difficult and I asked myself, do I go back to the REC (Research Ethics Committee) to let them know that this is an incidental finding, and I need to document it separately? But what we did is to go ahead and say to the parents that "look, we had set out to do this but the same test has also identified this, we didn't do an extra test, the same test has also recognized this" . . . we told them that this is different from what is normally in the population. We've done that." R24*

Respondents who did not share incidental findings with participants thought that since it was not planned for and no consent was obtained, then it was not necessary to share. Others felt that it would be done as aggregate results at the time of reporting findings from the planned study objectives.

*"So, we don't have to tell the people each and everything we are doing or investigating, we are giving them general findings. The consolidated findings where you draw graphs and they show in general, this is the response but those details where you get subgroups of people, we don't do that. It's also difficult to explain to the population. So we focus on our major objective but if there were some additional observations arising from the findings, those are research points." R021*

## Discussion

We set out to explore genetics and genomics researchers' perspectives and experiences regarding sharing incidental findings during the conduct of GGR in Uganda. Many of the respondents were agreeable to sharing incidental GGR findings with research participants giving a number of proposals on the nature of incidental findings and contexts/conditions under which incidental findings should be shared.

Emphasis was placed on sharing of beneficial, actionable, verified results, and results that are of public health interest. s Similar perceptions about sharing IFs have been documented where respondents from a population biobank study who were informed about an unexpected genetic finding evaluated this process as mainly positive. A cross-sectional, web-based survey investigated attitudes of 6944 individuals from 75 countries towards returning IFs from genome research and found that treatability and perceived utility of incidental results were deemed important with 98% of stakeholders [19, 20].

Genetic counselling was considered a crucial component of GGR. Counsellors are key players in facilitating participants' comprehension of complex genomics language, implications of receipt of results, risks and benefits involved. Other aspects to be clarified by counsellors include address misconceptions and allaying anxiety especially during communication of incidental findings. Hence the recommendation that sharing of incidental GGR results should be done in the presence of a genetic counsellor. However, despite the emphasis on the importance of genetic counselling, none of the respondents reported participation of qualified genetic counsellors in the respective genetics' studies. There is a capacity gap in this area. Our findings are similar to work reported in the USA where it has been observed that the genetic counsellors are important, their work is not well appreciated and may not be available for every research study [2, 21].

The need for regulation and ethical guidelines for conducting GGR including communicating incidental findings is key to setting ethical standards in practice. Although there are guidelines that regulate ethical conduct of research in Uganda, the existing guidelines do not address the unique challenges posed by GGR in this context [16]. Such context specific guidelines would provide a national framework for ethical oversight of GGR. The inadequacy of guidelines leaves unresolved the issues on whether or not to share incidental findings including what constitutes ethical practice. The need for ethical guidelines, including facilitation of research oversight by RECs, has been highlighted by other literature [15, 22]. Such oversight by RECs would mandate researchers to comply with the set standards and ethical obligation to "give back" to participants [3]. Although questions relating to how much, when and how information to give back including the fact that many have argued that there isn't really a general moral duty, especially given that returning results can sometimes be unhelpful or possibly even unwanted. Many of our respondents have on the contrary highlighted the fact that it is morally right and professionally necessary to share incidental findings particularly those deemed beneficial. For such sharing of findings to be appropriately conducted, the need for context specific national ethical guidelines for GGR is a growing reality [15].

Communication of incidental findings was considered acceptable provided informed consent had been sought. Such informed consent would best be obtained at the time of recruitment in the study; where participants are informed of the possibility of getting results that are outside the scope of the objectives of the current study. However, if consent was not obtained at recruitment, in case of incidental findings that are deemed beneficial, then re-consenting the participants should be done after seeking approval from the REC. This requirement for informed consent is an ethical standard for all research involving humans as participants. Although informed consent should ideally be obtained at recruitment, researchers are faced with a challenge of providing adequate information on incidental findings at the time of participant recruitment. Such dilemmas associated with informed consent for communicating incidental findings from genomic research have been highlighted by other scholars [7, 23]. The need to re-consent participants in order to share incidental genomics findings has also been discussed if such new information is to be released to research participants [14, 23].

A number of other ethical considerations for sharing incidental GGR findings were highlighted. These include proper evaluation of the risks and implications anticipated from sharing the results on the individual, family members and the entire community. This is attributed to the fact that genetic information though collected from a consenting individual, has potential to reveal information about the participant's family and sometimes the whole community. Release of such findings can cause anxiety and distress to individuals, cause family breakups or provoke widespread stigmatization of the community. Ethical issues including violation of privacy and breach of confidentiality associated with sharing incidental findings have been observed elsewhere [7]. Consent is also challenging in clinical genetic practice as

testing can generate 'incidental findings' with potential to reveal patient's family members' risk which may lead to breach of confidentiality [17]. Because of the potential risk associated with incidental findings, other commentators have discussed the consideration of the option of notifying participants about the right not to be re-contacted [24]. Feedback of incidental findings can sometimes be challenging if researchers discover sensitive information where test results could inadvertently reveal sensitive personal information particularly for vulnerable groups like children and fetuses. For example, during genetic testing for sickle cell disease, which is prevalent in Uganda, it's not uncommon to reveal discordant genetic information between the child and the male parent. And if not properly handled, such information could have dire consequences for the child and the family.

Although the majority of the respondents had not found incidental findings that they felt were worth sharing, some of those who had what they considered clinically significant incidental findings shared the results with their research participants and or the attending clinicians. However, owing to lack of clear ethical guidelines such sharing did not follow any systematic approach, procedures or standards. The decision and procedures were at the discretion of the researcher. As a result, researchers in such contexts experienced challenges. Issues to do with what constitutes appropriate ethical conduct, pre-conditions and procedures for sharing incidental findings. The different approaches employed in returning incidental findings namely, sending results to the attending clinicians, communicating to individual participants or their representatives and use of community meetings may have lacked adequate preparation and clear processes necessary for sharing sensitive unanticipated findings. This discrepancy in handling and sharing incidental findings is attributed to the lack of policy and specific ethical guidelines on how genomics testing results should be appropriately communicated. The challenge of lack of context specific evidence to inform guideline development in the African setting is also highlighted by the H3Africa guidelines for sharing genomics findings which recommends clinically valid and actionable results, but with an additional task to researchers to determine relevance of information for the target population. Additionally, many considered clinical benefit or results being actionable as the main premises for sharing incidental findings although determination of clinical utility of the results was also subjectively made by the researchers using no context relevant specified or regulated criteria. Sharing of incidental genomics findings based on clinical benefit and actionability has been highlighted in related works [3, 20, 21, 25].

Finally, there is limited literature that addresses GGR in the Uganda setting with none focusing on incidental genomic findings [1, 11, 13, 26]. The research reported in this study provided insight on the issues affecting feedback of incidental findings in a Ugandan setting that has not been documented before. Findings of this study and other related literature [1, 11, 13, 26] can go a long way in addressing the current gaps in GGR including data for development of ethics guidelines that are currently unavailable in Uganda. Additionally, other related settings particularly those on the African continent can adopt the findings for use in their context for example in places where stakeholders in research have considered feedback of GGR results important and expressed interest in return of results [27]. Our work to a good extent confirms what is proposed by international guidelines like the H3Africa guidelines published by the African Academy of Sciences with recommendations similar to what the Ugandan researchers propose, and this makes adoption of the international guidelines to the Ugandan setting feasible [16, 28].

## Limitations

We acknowledge the fact that there is a potential for social desirability bias that could make respondents to report favouring what they think to be societally preferred under the circumstances.

The research study being reported involved only researchers yet perceptions of other key stakeholders like research participants and research regulators would have provided a more complete picture on the views and experiences regarding genetics and genomics research findings with participants.

Finally, the fact that many of the respondents were known to the researchers might have put them in a situation where they felt obliged to participate.

## Supporting information

**S1 File.**
(DOCX)

**S2 File.**
(DOCX)

## Acknowledgments

We are grateful to the researchers who participated in this study.

## Author Contributions

**Conceptualization:** Joseph Ochieng, Betty Kwagala, Joseph Ali, Nelson K. Sewankambo.

**Data curation:** Joseph Ochieng, Betty Kwagala, John Barugahare, Deborah Ekusai-Sebatta.

**Formal analysis:** Joseph Ochieng, Betty Kwagala, Deborah Ekusai-Sebatta.

**Funding acquisition:** Joseph Ochieng, Betty Kwagala, John Barugahare, Erisa Mwaka, Joseph Ali, Nelson K. Sewankambo.

**Investigation:** Joseph Ochieng, John Barugahare.

**Methodology:** Joseph Ochieng, Betty Kwagala, John Barugahare, Erisa Mwaka, Joseph Ali.

**Project administration:** Joseph Ochieng.

**Resources:** Joseph Ochieng.

**Validation:** Joseph Ochieng.

**Writing – original draft:** Joseph Ochieng.

**Writing – review & editing:** Joseph Ochieng, Betty Kwagala, John Barugahare, Erisa Mwaka, Deborah Ekusai-Sebatta, Joseph Ali, Nelson K. Sewankambo.

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
