## [Decision Letter · Decision Letter 0]

24 Jan 2022

PONE-D-21-28187Perspectives and Experiences of Researchers Regarding Feedback of Incidental Genomic Research Findings: a qualitative studyPLOS ONE

Dear Dr. Ochieng,

Thank you for submitting your manuscript to PLOS ONE. After careful consideration, we feel that it has merit but does not fully meet PLOS ONE’s publication criteria as it currently stands. Therefore, we invite you to submit a revised version of the manuscript that addresses the points raised during the review process.

We look forward to receiving your revised manuscript.

Kind regards,

Vincenzo De Luca

Academic Editor

PLOS ONE

https://journals.plos.org/plosone/s/file?id=ba62/PLOSOne_formatting_sample_title_authors_affiliations.pdf”.

2. Please include additional information regarding the interview guide used in the study and ensure that you have provided sufficient details that others could replicate the analyses. For instance, if you developed an interview guide as part of this study and it is not under a copyright more restrictive than CC-BY, please include a copy, in both the original language and English, as Supporting Information.

“Research reported in this publication was supported by the National Human Genome Research Institute of the National Institutes of Health under Award Number U01HG009822. The content is solely the responsibility of the authors and does not necessarily represent the official views of the National Institutes of Health.”

4. Thank you for stating the following in the Funding Section of your manuscript:

“Research reported in this publication was supported by the National Human Genome Research Institute of the National Institutes of Health under Award Number U01HG009822. The content is solely the responsibility of the authors and does not necessarily represent the official views of the National Institutes of Health.”

“Research reported in this publication was supported by the National Human Genome Research Institute of the National Institutes of Health under Award Number U01HG009822. The content is solely the responsibility of the authors and does not necessarily represent the official views of the National Institutes of Health.”

Reviewers' comments:

Reviewer's Responses to Questions

**Comments to the Author**

1. Is the manuscript technically sound, and do the data support the conclusions?

Reviewer #1: Yes

2. Has the statistical analysis been performed appropriately and rigorously? 

Reviewer #1: I Don't Know

3. Have the authors made all data underlying the findings in their manuscript fully available?

Reviewer #1: Yes

4. Is the manuscript presented in an intelligible fashion and written in standard English?

Reviewer #1: Yes

5. Review Comments to the Author

Reviewer #1: The authors present a qualitative discussion of the perceptions, opinions, and (limited) experiences of Ugandan researchers on the subject of returning incidental findings to participants in genetic and genomic research projects. Their methodology is sound, and their findings are clearly presented. They provide many interesting and important relevant references. They could improve by explaining in the discussion exactly what new information this study brings to that discussion. As well, some specific suggestions or questions below.

Introduction: Might be worth mentioning that GGR has come leaps and bounds forward in the ten years since these references on incidental findings were published, and that an updated discussion on this topic is therefore much needed.

They highlight the specific need for expanding these discussions in African populations, which is important. But I’m not sure they adequately synthesize at the end how their findings might be important specifically for African researchers and African populations.

The sentence at lines 115-117 (reference 17) brings up a very interesting and complicated topic (about patient and family confidentiality in the context of genetic testing). Perhaps just one or two more sentences expanding on what is meant by this debate would be helpful to add.

They mention several times the possibility of genomic analysis uncovering an infectious disease or something that would be a public health concern. Could the authors expand on what such situations they envision, giving specific examples?

I am not sure about this sentence (at line 323-324): “unless it is a life-threatening event that one would error on the side of being wrong than right.” Would the possibility of reporting a life-threatening event that turned out to be an error not be severely damaging? I would think all results should have a high degree of certainty before being reported, and I felt the excerpts they include at this section support this as well.

6. PLOS authors have the option to publish the peer review history of their article (what does this mean?). If published, this will include your full peer review and any attached files.

Reviewer #1: **Yes: **Sarah Barclay

---

## [Author Response · Author response to Decision Letter 0]

6 Feb 2022

A point by point response to review comments document has been uploaded

---

## [Decision Letter · Decision Letter 1]

21 Mar 2022

PONE-D-21-28187R1Perspectives and Experiences of Researchers Regarding Feedback of Incidental Genomic Research Findings: a qualitative studyPLOS ONE

Dear Dr. Ochieng,

Thank you for submitting your manuscript to PLOS ONE. After careful consideration, we feel that it has merit but does not fully meet PLOS ONE’s publication criteria as it currently stands. Therefore, we invite you to submit a revised version of the manuscript that addresses the points raised during the review process.

We look forward to receiving your revised manuscript.

Kind regards,

Vincenzo De Luca

Academic Editor

PLOS ONE

Reviewers' comments:

Reviewer's Responses to Questions

**Comments to the Author**

1. If the authors have adequately addressed your comments raised in a previous round of review and you feel that this manuscript is now acceptable for publication, you may indicate that here to bypass the “Comments to the Author” section, enter your conflict of interest statement in the “Confidential to Editor” section, and submit your "Accept" recommendation.

Reviewer #2: (No Response)

2. Is the manuscript technically sound, and do the data support the conclusions?

Reviewer #2: (No Response)

3. Has the statistical analysis been performed appropriately and rigorously? 

Reviewer #2: (No Response)

4. Have the authors made all data underlying the findings in their manuscript fully available?

Reviewer #2: (No Response)

5. Is the manuscript presented in an intelligible fashion and written in standard English?

Reviewer #2: (No Response)

6. Review Comments to the Author

Reviewer #2: (No Response)

7. PLOS authors have the option to publish the peer review history of their article (what does this mean?). If published, this will include your full peer review and any attached files.

Reviewer #2: No

---

## [Author Response · Author response to Decision Letter 1]

24 Jul 2022

A point by point response to reviewer comments has been uploaded

---

## [Decision Letter · Decision Letter 2]

15 Aug 2022

Perspectives and Experiences of Researchers Regarding Feedback of Incidental Genomic Research Findings: a qualitative study

PONE-D-21-28187R2

Dear Dr. Ochieng,

We’re pleased to inform you that your manuscript has been judged scientifically suitable for publication and will be formally accepted for publication once it meets all outstanding technical requirements.

Kind regards,

Vincenzo De Luca

Academic Editor

PLOS ONE

Additional Editor Comments (optional):

Reviewers' comments:

Reviewer's Responses to Questions

**Comments to the Author**

1. If the authors have adequately addressed your comments raised in a previous round of review and you feel that this manuscript is now acceptable for publication, you may indicate that here to bypass the “Comments to the Author” section, enter your conflict of interest statement in the “Confidential to Editor” section, and submit your "Accept" recommendation.

Reviewer #2: All comments have been addressed

2. Is the manuscript technically sound, and do the data support the conclusions?

Reviewer #2: Yes

3. Has the statistical analysis been performed appropriately and rigorously? 

Reviewer #2: N/A

4. Have the authors made all data underlying the findings in their manuscript fully available?

Reviewer #2: Yes

5. Is the manuscript presented in an intelligible fashion and written in standard English?

Reviewer #2: Yes

6. Review Comments to the Author

Reviewer #2: Comments for the author:

This is a manuscript addressing researchers’ perspectives and experiences regarding feedback

of incidental genomics findings to participants performing a qualitative study on 30

respondents selected based on their role as genetics and genomics researchers in Uganda. They emphasized the absence of ethical standards and the unpredictability of the results. This paper brings up an interesting clinical issue. The authors of this work present an intriguing clinical problem, and they fully addressed the feedback. I have no other remarks to make about it.

7. PLOS authors have the option to publish the peer review history of their article (what does this mean?). If published, this will include your full peer review and any attached files.

Reviewer #2: No

---

## [Editor Report · Acceptance letter]

18 Aug 2022

PONE-D-21-28187R2 

Perspectives and Experiences of Researchers Regarding Feedback of Incidental Genomic Research Findings: a qualitative study 

Dear Dr. Ochieng:

I'm pleased to inform you that your manuscript has been deemed suitable for publication in PLOS ONE. Congratulations! Your manuscript is now with our production department. 

Kind regards, 

on behalf of

Dr. Vincenzo De Luca 

Academic Editor

PLOS ONE